# Polyimide-Based High-Performance Film Bulk Acoustic Resonator Humidity Sensor and Its Application in Real-Time Human Respiration Monitoring

**DOI:** 10.3390/mi13050758

**Published:** 2022-05-11

**Authors:** Yusi Zhu, Pan Xia, Jihang Liu, Zhen Fang, Lidong Du, Zhan Zhao

**Affiliations:** 1State Key Laboratory of Transducer Technology, Aerospace Information Research Institute, Chinese Academy of Sciences, Beijing 100190, China; zhuyusi16@mails.ucas.ac.cn (Y.Z.); ljh838841120@163.com (J.L.); zfang@mail.ie.ac.cn (Z.F.); 2School of Electronic, Electrical and Communication Engineering, University of Chinese Academy of Sciences, Beijing 100049, China; 3Personalized Management of Chronic Respiratory Disease, Chinese Academy of Medical Sciences, Beijing 100190, China

**Keywords:** film bulk acoustic resonator (FBAR), humidity sensor, dual-parameter detecting, polyimide (PI), respiration monitoring

## Abstract

Respiration monitoring is vital for human health assessment. Humidity sensing is a promising way to establish a relationship between human respiration and electrical signal. This paper presents a polyimide-based film bulk acoustic resonator (PI-FBAR) humidity sensor operating in resonant frequency and reflection coefficient S_11_ dual-parameter with high sensitivity and stability, and it is applied in real-time human respiration monitoring for the first time. Both these two parameters can be used to sense different breathing conditions, such as normal breathing and deep breathing, and breathing with different rates such as normal breathing, slow breathing, apnea, and fast breathing. Experimental results also indicate that the proposed humidity sensor has potential applications in predicting the fitness of individual and in the medical field for detecting body fluids loss and daily water intake warning. The respiratory rates measured by our proposed PI-FBAR humidity sensor operating in frequency mode and S_11_ mode have Pearson correlation of up to 0.975 and 0.982 with that measured by the clinical monitor, respectively. Bland–Altman method analysis results further revealed that both S_11_ and frequency response are in good agreement with clinical monitor. The proposed sensor combines the advantages of non-invasiveness, high sensitivity and high stability, and it has great potential in human health monitoring.

## 1. Introduction

Respiration is one of the vital signs of human beings because it can help assess an individual’s health condition and screen for early diseases based on information from respiratory rate and depth. Abrupt changes in respiration rate and depth often indicate the occurrence of illnesses, such as heart disease, bronchitis, asthma [1], pneumonia, chronic obstructive pulmonary disease (COPD) [2], sleep apnea syndrome (SAS) [3], and lung cancer. Therefore, respiration monitoring is helpful for the assessment of body condition and the medical diagnosis of diseases, especially in the case of the widespread spread of COVID-19. Conventional devices based on existing methods for respiration monitoring such as pulse oximetry [4], thermal sensors [5], and pressure sensors [6], are always uncomfortable, cumbersome and costly, which is inconvenient for patients. Thus, we need more appropriate ways for respiration monitoring.

Recently, humidity sensors with merits such as fast response, high accuracy, and reliability have been emerged as great candidates for the application in respiration monitoring, which is achieved by measuring changes in humidity during exhalation and inhalation. The drop in local relative humidity (RH) during inhalation, due to the air flow taking away the water vapor, and a subsequent rise in local relative humidity during exhalation due to the increased outflow of water vapor in the breath, happen during respiration, irrespective of the relative humidity of the environment. Therefore, relative humidity sensor-based respiration monitoring is less affected by the environmental conditions as compared to the mechanical sensors, conventional air flow meters and thermal sensors. Generally, there are various kinds of humidity sensors, such as optical fibers [7,8,9], capacitance [10,11,12,13,14], impedance [13,14,15,16,17,18], piezoelectric [19], surface acoustic wave (SAW) [20,21,22], and bulk acoustic wave (BAW) [23,24,25]. Among them, film bulk acoustic resonator (FBAR), a typical BAW, has gained wide attention in recent years due to their merits such as high sensitivity, high stability and fast response.

A lot of effort has been invested to realize high-performance FBAR humidity sensors. Hygroscopic materials such as polymers [26], graphene dioxide (GO) [27] and carbon nanomaterials are added to the surface of the resonators to increase the adsorption of water molecules and thus improve the sensitivity of the humidity sensors. Under the help of hygroscopic polymer polyvinylpyrrolidone (PVP) [26], the sensitivity was improved apparently for the thin PVP sample (−25 kHz/%RH) and thick PVP sample (−110 kHz/%RH) from 20%RH to 60%RH. Adding the GO as the sensitive layer of FBAR [27], the sensitivity was optimized to −5 kHz/%RH for low RH (3% < RH < 50%) and −25.5 kHz/%RH for high RH (70% < RH < 83%). Other methods, such as increasing the interaction path between water molecules and moisture-absorbing material ZnO by designing micro through holes in the top electrode [28], have also been proposed to improve the sensitivity of the sensor. The sensitivity of the proposed sensitivity-enhanced sensor was 3.2 times higher than that of a sensor with complete electrode. Although these above-mentioned methods have indeed improved the sensitivity of the FBAR humidity sensor to some extent, these methods are at the expense of the Q value of the resonator and increased the difficulty of the process, which poses difficulties for large scale fabrication. In addition, because the sensing mechanism of conventional FBAR is based on the frequency reduction caused by mass adsorption, it is inevitable that there are shortcomings such as low sensitivity at low humidity and nonlinear response at high humidity.

To solve the above-mentioned problems of the FBAR humidity sensor based on mass adsorption, our research group previously proposed a FBAR humidity sensor based on polyimide (PI) Young’s modulus modulation [24] rather than the traditional gravimetric response. The PI film functioned as a support structure and a humidity sensitive structure simultaneously. As the support layer, PI film makes the FBAR structure more robust and simplifies the manufacturing process. As the humidity sensitive layer, PI enhanced the humidity sensitivity to +67.3 kHz/%RH (64 ppm/%) between 15%RH and 85%RH with a sufficient Q (>200), which is 39 times higher than the sample without PI film. Inspired by the improved sensor performance by two-parameter detection, such as resonant frequency and resistance [29], resonant frequency and reflection coefficient S_11_ [30], we further investigated the reflection coefficient S_11_ response of the PI-FBAR humidity sensor [25]. Results showed that an ultra-high average humidity sensitivity of 0.044 dB/%RH (2095 ppm/%RH) was achieved, and the sensitivity at 20% low relative humidity reached 0.075 dB/%RH (3571 ppm/%RH), which is more than two orders of magnitude higher than that of the resonant frequency response. However, the humidity response mechanism of S_11_ has not been systematically studied. In addition, as far as we know, there is no report on human respiration monitoring based on FBAR humidity sensor, especially the new detection method of S_11_ parameter response for respiration monitoring.

In this work, a PI-based FBAR humidity sensor operating in resonant frequency and reflection coefficient S_11_ dual-parameter sensing was analyzed, designed, fabricated, and tested. The application of PI-based high-performance FBAR humidity sensor in real-time respiratory monitoring was performed. The responses of both the two parameters of resonant frequency and S_11_ to respiratory monitoring were experimented and compared. Both parameters were used to sense and monitor different rates of breathing, such as normal breathing, apnea, slow breathing, and fast breathing, as well as to sense the difference between oral and nasal breathing. Besides, a comparative analysis of the respiratory characteristic curves before and after exercise was carried out to predict the fitness of individual, especially for athletes. The respiration signals before and after water drinking were also well recorded and systematically analyzed for daily water intake warning. In order to prove the accuracy of the proposed PI-FBAR humidity sensor in human respiratory monitoring, respiratory rate test results of our prepared sensor operating in resonant frequency and S_11_ dual-parameter were compared with that measured by a clinical monitor.

## 2. Materials and Methods

### 2.1. Analysis and Simulation of S_11_ Response

Though, it has been verified in our previous work [25] that the resonant frequency response is based on the modulation of the humidity on the Young’s modulus of PI, the humidity response mechanism of S_11_ is not clear, and it is tentatively presumed to be caused by the change in acoustic loss caused by water molecules. To explore the humidity sensing mechanism of S_11_ more systematically, Mason equivalent transmission line model [31] is used to analyze and simulate the PI-FBAR. The Mason equivalent transmission line model of PI-FBAR multilayer structure is shown in Figure 1, which contains general acoustic layers, such as top/bottom Pt electrode layers and support/sensitive PI layer, and ZnO piezoelectric acoustic layer, which adds electromechanical coupling on the basis of general acoustic layer.

Parameters in the above Mason model are defined as follows:(1)$$\left\{\begin{array}{l} a_{n}=Z_{n}\tan(\frac{k_{n}d_{n}}{2})\\ b_{n}=\frac{Z_{n}}{j\sin(k_{n}d_{n})} \end{array}\right.(n=1,2,3\ldots )$$
where $Z_{n}$, $k_{n}$, and $d_{n}$ are acoustic impedance, wave number, and thickness of each layer, respectively.

The acoustic impedance $Z_{n}$ and wave number $k_{n}$ can be expressed as:(2)$$\left\{\begin{array}{l} Z_{n}=\sqrt{\rho c_{33}^{E}}\\ k_{n}=2\pi f\sqrt{\frac{\rho }{c_{33}^{E}}}-j\alpha \end{array}\right.$$

As can be seen in Equation (2), the properties of PI-FBAR are related to the elasticity coefficient $c_{33}^{E}$, density ρ, and attenuation factor α of the material.

Other parameters in Mason model are as follows: $C_{0}=\varepsilon_{zz}^{s}/d$ presents the static capacitance, and $hC_{0}$ is the transformer turns.

Given the thickness and material parameters of each layer of PI-FBAR, as shown in Table 1, we can perform S-parameter simulation analysis on PI-FBAR through ADS (Advanced Design System, ADS) RF circuit simulation software.

As we all know, PI is a porous polymer material, whose viscoelasticity will change with the change in humidity [32]. As the path of acoustic wave propagation, when the acoustic wave is transmitted to the PI layer, the change in the attenuation factor of PI will cause the change in S_11_ value of the resonator. In order to explore the modulation effect of attenuation factor α on S_11_, we simulated and analyzed the relationship between S_11_ and the change in α using ADS with control variable method. The simulation results are shown in Figure 2, as the α increases from 30,000 to 40,000, which presents the increased loss caused by the increase in viscoelasticity, the resonant frequency of the resonator remains unchanged, but the value of S_11_ increases from −32 dB to −30 dB. Although the related expression cannot be given due to the complex relationship between the attenuation factor and humidity, the simulation results show that the increased PI viscoelasticity caused by the increase in humidity will lead to an increase in the value of S_11_.

### 2.2. Fabrication of PI-Based FBAR Humidity Sensor

The designed PI-FBAR sensor chip was fabricated with the MEMS process described in our previous works [24,25], including: deposition of SiO_2_ sacrifice and insulation layer, deposition and patterning of Pt bottom electrode, magnetron sputtering and patterning of ZnO piezoelectric film, deposition and patterning of Pt top electrode, spin coating and patterning PI layer, and etching SiO_2_ and Si from the back side. After the proposed sensor chip was prepared, the chip was fixed to the PCB board so that the sensor chip and the PCB board can be electrically interconnected by the gold wire ball welding technology, and then connected to the vector network analyzer through the SMA interface for subsequent testing.

The preparation of PI film is crucial because it acts as both the supporting structure and the humidity sensitive structure. The polyamide acid solution used for the preparation of PI film is ZKPI-305IIE, purchased from Beijing POME Technology Co., LTD (Beijing, China). The specific preparation process of PI is shown in Figure 3, including the following steps: (1) Cleaning substrate with 100 W of oxygen plasma for 3 min to enhance the adhesion of the polyamide acid solution to the substrate; (2) Spin coating the polyamide acid solution on the wafer at the speed of 3000 r/min to obtain a subsequent polyimide film with a thickness of about 4 um; (3) Pre-baking the wafer on a heating plate at 100 °C for 40 min to evaporate most of the water; (4) The polyamide acid is completely iminoized into a polyimide film in a programmable heating oven by step heating method: 150 °C/60 min, 180 °C/30 min, 250 °C/60 min, and 300 °C/30 min.

### 2.3. Apparatus for Humidity Measurement and Breath Monitoring

As discussed in our previous works [24,25], due to the loose and porous structure of the PI film and containing hydrophilic functional groups, water molecules can diffuse into the PI film easily during the physical adsorption of water molecules. On the one hand, the Young’s modulus and density of PI can be modulated by the absorption and desorption of water moisture, which in turn changes the resonant frequency of the PI-FBAR. On the other hand, as the path of acoustic wave propagation, the change in viscoelasticity of PI will also cause the change in loss of the resonator when it resonates, thereby changing the reflection coefficient S_11_ of the resonator. Specifically, as the ambient humidity increases, the PI absorbs moisture, which increases the resonant frequency and the S_11_ value. As the measured ambient humidity decreases, PI desorbs moisture, which reduces the resonant frequency and the S_11_ value.

The sensing performance test system of the PI-FBAR humidity sensor is shown in Figure 4. As shown in Table 2, different saturated salt solutions in equilibrium state produce different humidity environments. Specifically, saturated salt solution with a relative humidity of 50% was prepared from a mixed solution of K_2_CO_3_ and KI. The proposed humidity sensor was inserted into a closed glass bottle and placed above the level of a saturated salt solution. A vector network analyzer (VNA, E5061B network analyzer) combined with the LabVIEW program was used to measure and record the real-time change in the resonant frequency and S_11_.

The diagram of the real-time human breathing monitoring system is shown in Figure 5. The tested volunteer was required to wear mask, which contained our fabricated humidity sensor. The sensor was connected to the VNA through a high-frequency line, and the respiratory output signal waveforms were controlled and recorded in real time through the LabVIEW program. Further, to prove the accuracy of the proposed PI-FBAR humidity sensor in human respiratory monitoring, we compared the test results of the sensor’s resonant frequency and S_11_ responses of the respiratory rate with that measured by a clinical monitor (BeneView T5, Mindray, Shenzhen, China), which obtains the breathing signal from the impedance change caused by human breathing.

## 3. Results and Discussion

### 3.1. Structure Characterization

Figure 6a presents the top SEM image of the as-prepared PI film, from which we can see that the PI film is porous as we discussed above. Such a porous structure is beneficial to the adsorption of water molecules. Figure 6b,c are the cross-section SEM images of the ZnO piezoelectric film and its XRD (X-ray diffraction, XRD) pattern, respectively. Results showed that the piezoelectric material has a thickness of 1.48 µm with good verticality, indicating its good piezoelectric properties. Other structural parameters and properties can be seen in our previous work [25]. The small difference in fundamental resonance frequency and reflection coefficient S_11_ from those reported in previous work may be due to the slight difference in the thickness of ZnO during magnetron sputtering and the difference in the surface properties of PI film.

### 3.2. Humidity Response of PI-FBAR Sensor

The prepared PI-FBAR humidity sensor was placed into different saturated salt solutions in equilibrium state with different relative humidity (RH) to test its humidity response characteristics. It should be pointed out that the sealed bottle with desiccant is regarded as 0%RH. Keep the humidity sensor in an airtight container with desiccant for 120 s, of which the RH is regarded as 0%, then quickly place it in a saturated salt solution with certain RH for 40 s, then place it in an airtight container with desiccant for 120 s, repeat this operation, until these different humidity environments of 11%, 33%, 50%, 75%, and 98% are repeated. Specifically, due to the large hysteresis from a high humidity environment of 98% to a dry environment of 0%, this process lasted for 3 min to make the sensor as pristine as possible. In order to test the continuous response stability of the humidity sensor, we conducted three rounds of experiments. The humidity response results are shown in Figure 7a. It can be seen from the response results that the frequency response shows a larger baseline drift than the S_11_ response, which may be caused by the fact that the frequency response requires a longer time to absorb and desorb humidity, and the frequency response is less stable than the S_11_ response. The baseline drift would be improved if the latency of the test in Figure 7a were lengthened.

The humidity response sensitivity (S) of resonant frequency and S_11_, calculated by Equation (3), were shown in Figure 7b.
(3)$$\left\{\begin{array}{l} S_{S_{11}}=\frac{\Delta S_{11}}{\Delta RH}/S_{11}\\ S_{f}=\frac{\Delta f}{\Delta RH}/f \end{array}\right.$$

Both the resonant frequency and the S_11_ response sensitivity showed higher linear sensitivity at low humidity and lower linear sensitivity at high humidity. The reduced sensitivity in high humidity environment is mainly caused by the PI mass loading effect in high humidity environment, which reduced the frequency of the resonator, thereby offsetting the value of the frequency increase caused by the increase in PI’s Young’s modulus. For the lower relative humidity from 0% to 33%, the sensitivity of the resonant frequency and the S_11_ response are 56 ppm/RH and 964 ppm/RH, respectively, and for the higher relative humidity from 33% to 98%, the sensitivity of the resonance frequency and the S_11_ response are 15 ppm/RH and 570 ppm/RH. Although the sensitivities of both parameters are reduced in higher relative humidity environments, they are still sufficient for subsequent human respiration monitoring applications. Results showed that the sensitivities of S_11_ in both high and low humidity environments are several tens of times higher than those of resonant frequency.

Besides, in order to analyze the stability of the sensor more intuitively, we compared the response results of three rounds of experiments, as shown in Figure 7b. The results showed that the resonant frequency and S_11_ response have good repeatability, and the stability of S_11_ is even better than that of the resonant frequency response.

Apart from that, the sensor’s stability was tested by measuring the resonant frequency and S_11_ responses for 6 h in different humid environments (11%, 33%, 50%, and 75%RH). As shown in Figure 8a for frequency while Figure 8b for S_11_ response, respectively, these two responses showed that the proposed sensor possesses high stability with small errors.

The real-time responses of the resonant frequency and S_11_ with periodic changes in humidity from 0 to 98% were shown in Figure 9a, and the results showed that both of these two variables have very good stability. Curves of adsorption response time at RH from 0% to 98% and desorption response time at RH from 98% to 0% were illustrated in Figure 9b. For the S_11_ response, the time of 63% and 90% complete adsorption response are 6 s and 15 s, respectively, and those of desorption response are 7 s and 19 s, respectively. For the resonant frequency response, the time of 63% and 90% complete adsorption response are 5 s and 12 s, respectively, and those of desorption response are 9 s and 26 s, respectively. These results showed that the resonant frequency response is faster than that of S_11_ in the process of moisture absorption, while the response of S_11_ is faster than that of the resonant frequency in the process of desorption. The humidity response performance of both parameters is sufficient for most humidity detection applications. In addition, study [33] has shown that the test temperature will affect the adsorption and desorption response times of the humidity sensor. When the temperature increased from 25 °C to 40 °C, the adsorption and desorption response times decreased to varying degrees, and specifically, the desorption time was shortened by half. Therefore, when the PI-FBAR humidity sensor is used for human respiration monitoring, the response speed of the sensor should be faster due to the increased temperature of the exhaled air, which is close to 37 °C of the human body, and the promotion effect of the respiratory airflow.

Comparisons of the performance of our designed sensor in this work with some other piezoelectric acoustic resonant humidity sensors in previous literature are listed in Table 3. Compared to other humidity sensors based on FBAR or SAW, our designed PI-FBAR sensor presented significantly higher sensitivity as well as remarkable linearity.

### 3.3. Respiration Monitoring Using Humidity Sensor

Monitoring of different breathing conditions was performed on a healthy adult volunteer to study the feasibility of the PI-FBAR humidity sensor performance (see Appendix A). During the breathing cycle, the sensor was exposed to varying levels of humidity in the inhaled and exhaled breathing. It should be pointed out that to better analyze the respiratory signal, a band-pass filter with a cut-off frequency of 0.05–0.8 Hz was used to remove baseline drift and high-frequency noise. The processed signal is equal to the band-pass filtered signal plus the mean of the original signal. These observed resonant frequency and reflection coefficient S_11_ curves, shown in Figure 10b, indicated an increase in the frequency and S_11_ during expiratory periods, while there was a decrease during the inspiratory periods. This can be attributed to the fact that a large amount of water vapor is brought in when exhaling, causing the humidity to rise, while most of the water vapor is taken away due to the air flow when inhaling, making the humidity decrease, which is consistent with the sensor’s frequency and S_11_ response of humidity as described above. The results shown in Figure 10a also reveal that the shift of the resonant frequency and S_11_ parameters caused by deep breathing are greater than that of normal breathing, which may be attributed to the significantly large volume of air movement during the deep breathing process. Additionally, deep breathing is usually accompanied by a decrease in respiratory rate (RR). These significant differences between normal and deep breathing suggested the applicability of the fabricated PI-FBAR humidity sensor for identification and diagnosis of several health conditions such as asthma, tachypnoea, etc.

Performance of the fabricated sensor under different breathing modes, such as nasal breathing and oral breathing, was also analyzed. As shown in Figure 11a, the resonant frequency shift and S_11_ variation during oral breathing are greater than that of nasal breathing. The frequency shift of nearly 0.4 MHz and a change in S_11_ value nearly 0.2 dB were observed during the nasal breathing, whereas a frequency shift of nearly 0.6 MHz and a change in S_11_ value nearly 0.3 dB were observed during the oral breathing. The increase in the changes in frequency and S_11_ in case of oral breath is attributed to the large intake of air volume and subsequent increase in number of molecules interacting with the sensor as compared to that of the nasal breath. Furthermore, the presence of nasal mucosa creates filtered air which may also affect the overall humidity during the nasal breathing mode.

The fabricated sensor was thereafter used for continuous inspection on respiration rate (RR). For normal adults, the normal breathing rate is 12–20 per minute, and breathing rates outside the normal range may indicate health problems. The response curves of S_11_ and frequency of the humidity sensor for nasal breathing with different respiration rates are demonstrated in Figure 11b. The RR can be calculated by counting the number of peaks of the captured cyclic signal in a measured interval of time, usually within 60 s.

The respiration rates for slow, normal, and fast breath modes observed in Figure 11b are 7, 17, and 31 per minute, respectively. Results showed that both the resonant frequency and reflection coefficient S_11_ can efficiently distinguish between varying breathing rates, indicating their potential for real time health monitoring.

The respiration rate (RR) and depth of respiration (DR) before and after climbing six floors were also monitored and breathing index (BI) was calculated to estimate the fitness level of five individuals. As can be observed from Figure 12a,b, when subject 2 switched from rest to a vigorous exercise, both the RR and DR changes. DR is calculated based on the S_11_ or frequency change in every respiration state. In case of nasal breathing before vigorous exercise, as shown in Figure 12a, the RR and DR calculated from S_11_ response are 16 per minute and nearly 0.15 dB, and those calculated from resonant frequency response are 16 per minute and nearly 0.32 MHz. However, in the case of nasal breathing after vigorous exercise, as shown in Figure 12b, the RR and DR calculated from S_11_ response change to be 36 per minute and nearly 0.11 dB, and those calculated from resonant frequency response are 36 per minute and nearly 0.22 MHz.

The empirical metric, Breathing Index (BI), is defined as the ratio of respiration rate to the depth of respiration under light and vigorous exercises [34]. In our experiment, the rest state before climbing six floors refers to light exercise, and climbing six floors refers to vigorous exercise. A subject is said to be fit if the BI is close to 1, which can be represented mathematically as follows:(4)$$BI=\left(\frac{DR_{climbing}}{DR_{rest}}\right)\times \left(\frac{RR_{climbing}}{RR_{rest}}\right)$$
where $DR_{climbing}$ and represent the depth of respiration and respiration rate after vigorous, while $DR_{rest}$ and $RR_{rest}$ represent the depth of respiration and respiration rate at rest state. 

By using Equation (4), the BI of subject 2 at frequency response and S_11_ response were calculated to be approximately 1.55 (>1) and 1.65 (>1), respectively. Results (BI > 1) revealed that subject 2 exchanged a larger volume of air during climbing six floors than he did during rest state. The calculated BIs of the five subjects were shown in Figure 12c, analysis results indicated that subject 3 and 4 in our experiment are more fit than subject 2 because they both had a smaller BI index. The same is true in reality; subjects 3 and 4 swam or exercised about 5 times a week, while subject 2 only exercised almost once a week. In any case, all five subjects in our experiment were fitter than the subject in reference [35], since the BI calculated of subject in our experiment is much smaller than that of subject in reference [35], which reached 3.37. Results showed that both the resonant frequency and reflection coefficient S_11_ can be used to efficiently predict the fitness of individual, especially for athletes.

As one of the indicators of personal health, daily water intake can also affect respiratory humidity. Respiration signals of the proposed PI-FBAR sensor were also recorded before and after water intake, and the dynamic response curves of nasal breathing were plotted in Figure 13. The subject was asked not to drink water one hour before the test. Results showed that both the S_11_ and resonant frequency increased after water intake, and then showed a tendency to decrease over time, indicating the process of water loss. Results proved that our sensors have the potential application in the medical field for detecting the loss of body fluids and daily water intake warning.

In summary, our proposed humidity sensor can not only record the user’s respiration rate and respiration manner, but also predict the fitness and monitor the dehydration of individual, which is useful for patients’ clinical nursing. More notably, from the monitored signal curves, the response of S_11_ parameters to different breathing modes and breathing states is better than that of resonant frequency, which is manifested in the smaller glitch of the waveform and more balanced amplitude, indicating that the S_11_ parameter is more suitable for respiration monitoring.

In order to further prove the accuracy of the proposed PI-FBAR humidity sensor in human respiratory monitoring, we compared the respiratory rate test results of the sensor’s resonant frequency and S_11_ responses with that measured by the clinical monitor. The experimental comparison results are shown in Figure 14. As shown in Figure 14a,b, the Pearson correlations between the S_11_ and resonant frequency response results with that of the clinical monitor, in the range of different respiration rates ranging from a few to three dozen, were 0.982 and 0.975, respectively. The highly correlated results with clinical monitor demonstrated the reliability of our fabricated sensor for respiratory monitoring.

In addition, the consistency of S_11_ and resonant frequency response results with that of the clinical monitor were also analysed by Bland–Altman method [36]. Its basic idea is to calculate the 95% consistency limit of the difference between the two measurement results. If this limit is within the clinically acceptable range, it can be considered that the two measurement methods have good consistency and can be replaced by each other. As shown in Figure 14c,d, the 95% consistency limit of the difference between the respiration rate measured by S_11_ and frequency response with that measured by clinical monitor were (−2.870, 2.525) and (−3.386, 2.961), respectively. The results showed that most of the differences fall within the 95% confidence interval. After analysing the original breathing signal, it can be seen that very few points with larger differences are due to the sudden change in the breathing rate. Different breathing rate extraction algorithms lead to inconsistency in the calculation results of the breathing rate. Anyway, results showed that both S_11_ and frequency response are in good agreement with clinical monitor, and the results measured by S_11_ had better consistency.

## 4. Conclusions

In this work, a polyimide-based film bulk acoustic resonator (PI-FBAR) humidity sensor operating in resonant frequency and reflection coefficient S_11_ dual-parameter was proposed and fabricated for real-time human respiration monitoring. The proposed PI-FBAR sensor operating in resonant frequency was governed by the modulation of the humidity on the Young’s modulus of PI, which is different from the conventional FBAR governed by mass loading effect, whereas the proposed PI-FBAR sensor operating in S_11_ was based on the modulation of the humidity on attenuation factor of PI layer. Humidity response test results showed that both resonant frequency and S_11_ dual parameters exhibit high sensitivity and stability, and particularly, the sensitivity of the S_11_ mode was dozens of times higher than that of the frequency. The prepared dual-parameter PI-FBAR, especially that operating in S_11_ mode, was proposed for the first time for human respiration monitoring. Results showed that both the frequency shift and reflection coefficient S_11_ variation can be used to sense different breathing conditions, such as normal breathing and deep breathing, normal breathing, slow breathing, apnea, and fast breathing by detecting respiration rate and depth of respiration. They can also be used for detection in different breathing modes, such as nasal breathing and mouth breathing. Experimental results also indicated that the proposed humidity sensor had potential applications in predicting the fitness of individual and in the medical field for detecting body fluids loss and daily water intake warning. The respiratory rate measured by our proposed PI-FBAR humidity sensor operating in frequency mode and S_11_ mode has a Pearson correlation of up to 0.975 and 0.982 with the respiratory rate measured by the clinical monitor, respectively. Bland–Altman method analysis results further showed that both S_11_ and frequency response are in good agreement with clinical monitor, and the results measured by S_11_ had better consistency. In summary, our proposed sensor combines the advantages of non-invasiveness, high sensitivity and high stability, and holds tremendous potential in human health monitoring.

## Figures and Tables

**Figure 1 micromachines-13-00758-f001:**
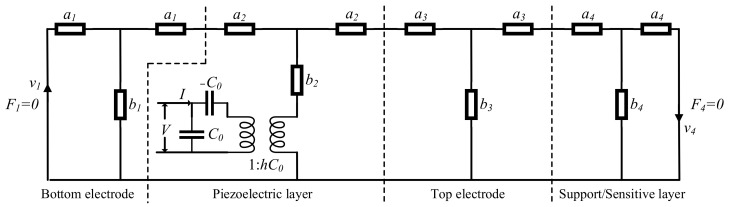
Mason equivalent transmission line model of PI-FBAR multilayer structure.

**Figure 2 micromachines-13-00758-f002:**
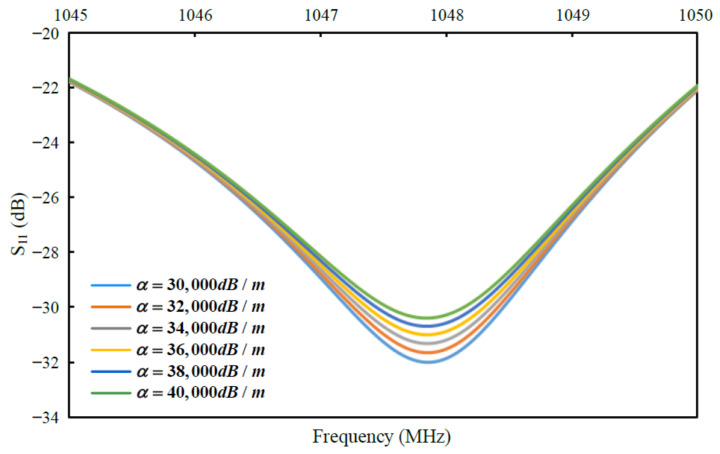
Simulation results of the response of S_11_ with the change in attenuation factor α.

**Figure 3 micromachines-13-00758-f003:**

Preparation procedures of PI film, including: (**a**) substrate cleaning; (**b**) spin coating polyamide acid solution; (**c**) pre-baking; (**d**) imidation by step heating method.

**Figure 4 micromachines-13-00758-f004:**
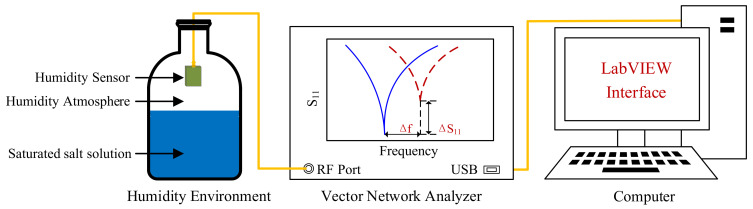
Measurement system for humidity sensing. Reproduced with permission from [25]. Copyright © 2021, IEEE.

**Figure 5 micromachines-13-00758-f005:**
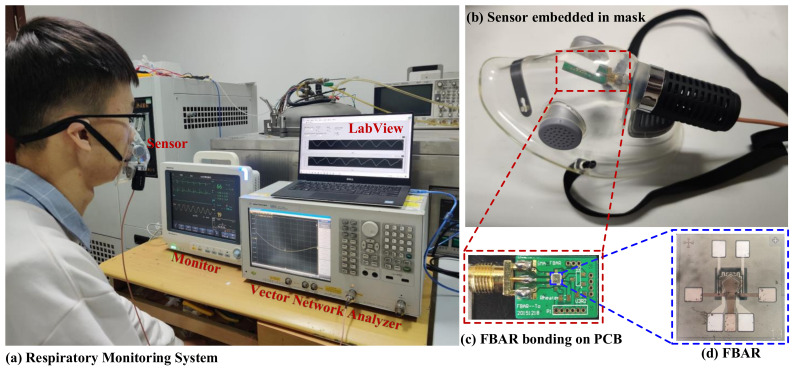
(**a**) Respiration monitoring system; (**b**) PI-FBAR humidity sensor embedded in breathing mask; (**c**) PI-FBAR bonding on PCB; (**d**) image of PI-FBAR.

**Figure 6 micromachines-13-00758-f006:**
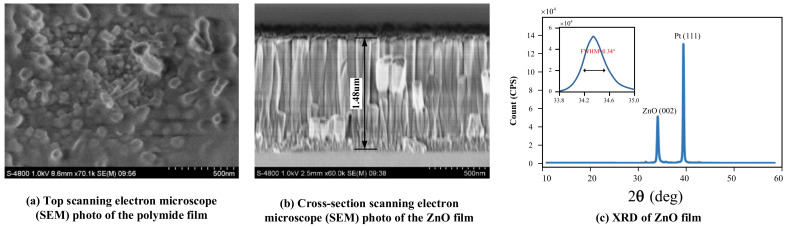
(**a**) Top scanning electron microscope (SEM) photo of PI film; (**b**) Cross-section SEM of ZnO piezoelectric film; (**c**) XRD pattern of ZnO film.

**Figure 7 micromachines-13-00758-f007:**
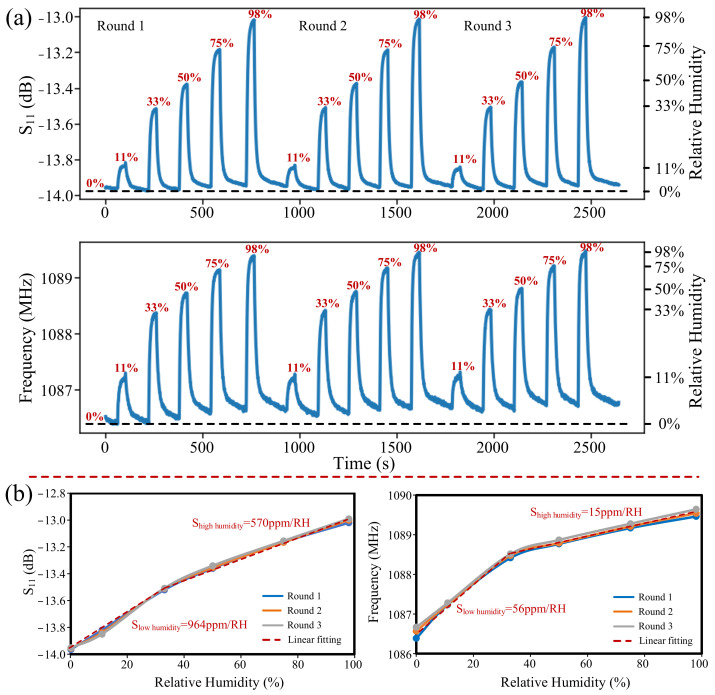
(**a**) Humidity responses of S_11_ and resonant frequency in the RH range of 0% to 98% for three rounds; (**b**) Sensitivity and stability curves of S_11_ and resonant frequency humidity responses.

**Figure 8 micromachines-13-00758-f008:**
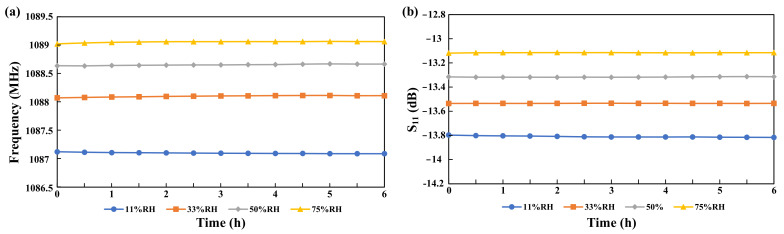
Stability analysis of the sensor for 6 h (**a**) for resonant frequency and (**b**) for S_11_ in different humidity environments.

**Figure 9 micromachines-13-00758-f009:**
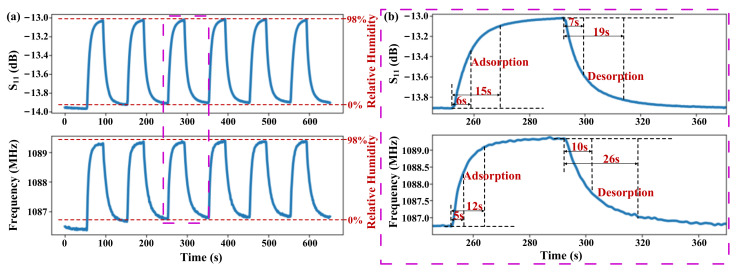
(**a**) Real-time response and (**b**) Humidity response time and recovery time performance of the resonant frequency and S_11_ with periodic changes in humidity from 0% to 98% and 98% to 0%.

**Figure 10 micromachines-13-00758-f010:**
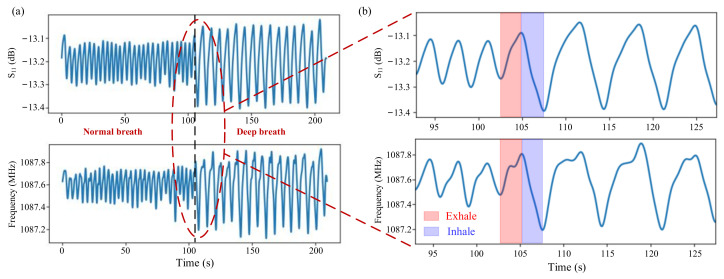
(**a**) S_11_ and frequency responses with breathing time for normal breathing condition and deep breathing condition; (**b**) Partial enlargement of (**a**).

**Figure 11 micromachines-13-00758-f011:**
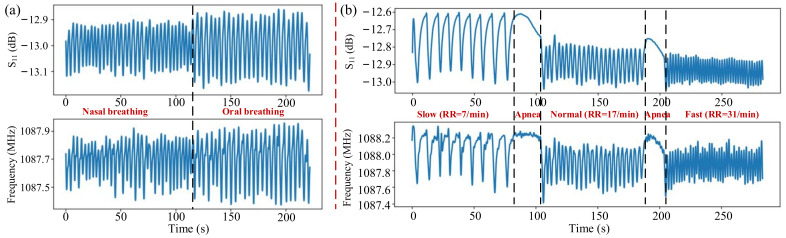
(**a**) S_11_ and frequency responses with breathing time in case of nasal breathing and oral breathing modes and (**b**) nasal breathing with different breathing rates: slow breath, apnea, normal breath, and fast breath.

**Figure 12 micromachines-13-00758-f012:**
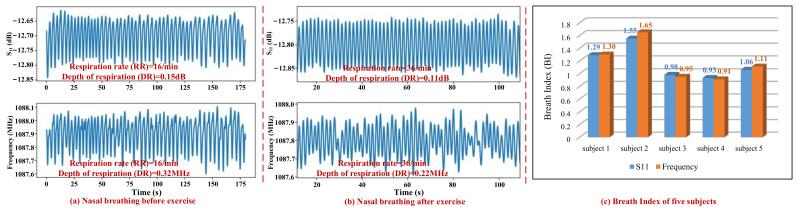
S_11_ and frequency responses of subject 2 with nasal breathing (**a**) before and (**b**) after climbing six floors; (**c**) Calculated Breath Index (BI) of five subjects.

**Figure 13 micromachines-13-00758-f013:**
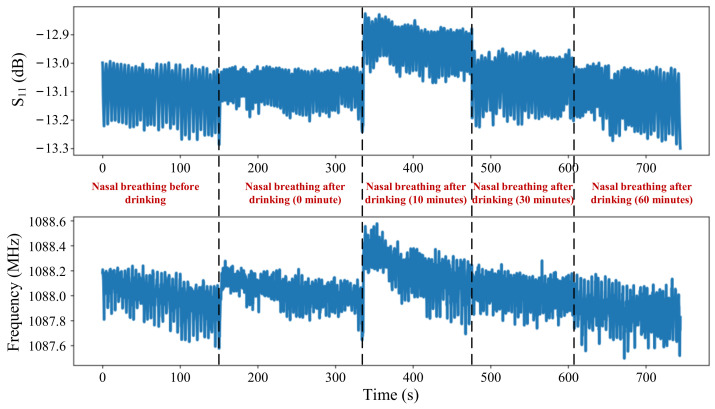
S_11_ and frequency responses with nasal breathing time before water intake and after water intake for 0 min (immediately), 20 min, 40 min and 60 min.

**Figure 14 micromachines-13-00758-f014:**
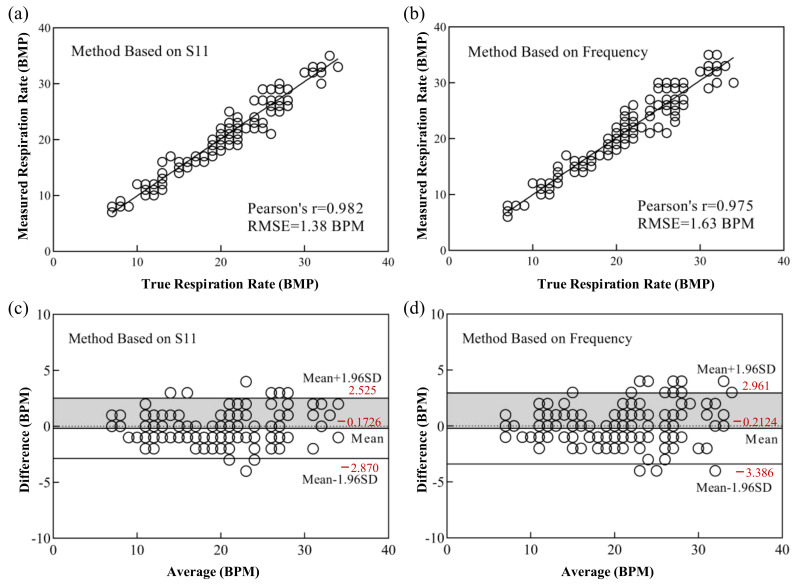
Correlation plots of (**a**) S_11_ and (**b**) frequency with clinical monitor test results. Bland–Altman plots of (**c**) S_11_ and (**d**) frequency with clinical monitor test results.

**Table 1 micromachines-13-00758-t001:** The thickness and material parameters of each layer in the PI-FBAR multilayer film equivalent transmission line model.

Layer	Material	Thickness d (um)	Elasticity Coefficient $c_{33}^{E}$ (GPa)	Density ρ (kg/m^3^)	$\mathrm{Dielectric}\;\mathrm{Constant}\;\varepsilon_{zz}^{S}\;(10^{-11})$	Longitudinal Sound Velocity v (m/s)	$\mathrm{Acoustic}\;\mathrm{Impedance}\;Z_{n}\;(10^{7}\mathrm{kg}/m^{2}s)$	Attenuation Factor α (dB/m)
Bottom electrode	Pt	0.2	168	21,450	/	2800	6.00	17,760
Piezoelectric layer	ZnO	1.5	211	5600	7.79	6340	3.62	2500
Top electrode	Pt	0.2	168	21,450	/	2800	6.00	17,760
Support/Sensitive layer	PI	4.2	5.4	1300	/	2038	0.26	30,000

**Table 2 micromachines-13-00758-t002:** The relative humidity of different saturated salt solutions (25 °C).

Saturated Salt Solution	Relative Humidity (%RH)
LiCl	11
MgCl_2_	33
K_2_CO_3_	43
KI	70
NaCl	75
K_2_SO_4_	98

**Table 3 micromachines-13-00758-t003:** Comparison of the performance of piezoelectric resonant humidity sensors in previous literatures.

Reference	Device Type	Resonant Frequency (MHz)	Sensitive Material	RH Range (%RH)	Sensitivity (ppm/%RH)	Linear
[26]	FBAR	700	PVP	20–70	−157	No
[27]	FBAR	1250	GO	3–7070–80	−4 −20.4	No
[28]	FBAR	1265	ZnO	25–88	−9.2	No
[24]	FBAR	1055	PI	15–85	+63.8	Yes
[25]	FBAR	1048	PI	20–80	+28.56 (frequency mode) +2078 (S_11_ response)	Yes No
[20]	SAW	392	GO	10–90	−107	No
[22]	SAW	202	3DAG/PVA/SiO_2_	5–5555–90	−4.96−12.1	No
This work	FBAR	1089	PI	11–33	+56 (frequency mode) +965 (S_11_ response)	Yes
				33–98	+15 (frequency mode) +570 (S_11_ response)	Yes

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
