# Peer review of "Polyimide-Based High-Performance Film Bulk Acoustic Resonator Humidity Sensor and Its Application in Real-Time Human Respiration Monitoring"

_micromachines, 2022, doi:10.3390/mi13050758_

Round 1

Reviewer 1 Report

In this paper, the authors reported a FBAR humidity sensor for respiration monitoring. The idea is interesting, the text is well arranged, and the logic is clear. So I recommend that this manuscript can be accepted. The authors should avoid using abbreviation in the title. The references format needs to be carefully revised to fit the journal requirement. Some of current reference format is not consistent with each other. Please double check the reference format, as there are lots of mistakes in journal volume/page number, uppercase/lowercase, hybrid using of journal full name and abbreviation.

Reviewer 2 Report

This work presents a systematic study of a PI-FBAR humidity sensor operating in frequency mode and S11 mode, and demonstrates its applications in high accuracy human respiration monitoring. The manuscript is well organized and the experiments are also systematically conducted. It can be published after the following minor issues are addressed:

  1. Some typing mistakes should be corrected, such as line 96: “sensing was analyzed, analyzed....”
  2. In line 138, the author mentioned that “PI is a porous polymer material, whose viscoelasticity will change with the change of humidity”. References should be provided to make readers understand this knowledge easily.
  3. Line 192, the detailed information of the saturated salt solutions (such as materials, concentrations) should be added, or provide the related references.
  4. In figure 7a, the frequency response shows a big baseline drift compared to that of the S11 response, what is the reason for such difference? A brief explanation should be added.
  5. Line 270-272: “the resonant frequency response is faster than that of S11 in the process of moisture absorption, while the response of S11 is faster than that of the resonant frequency in the process of desorption.” A brief explanation for this phenomenon should be added.
  6. Figure 4 has some overlap with the author’s previous publication, it is better to remake this figure to avoid the copyright issue.
  7. Font size in some figures is too small, such as Figure 8 (7s, 19s), and Figure 11 (Respiration rate......). 

Reviewer 3 Report

Reviewer coments:

This paper presents a polyimide-based film bulk acoustic resonator (PI-FBAR) humidity sensor operating in resonant frequency and reflection coefficient S11 dual-parameter with high sensitivity and stability, and it is applied in real-time human respiration monitoring.  The proposed sensor combines the advantages of non-invasive, high sensitivity, high stability, and has great potential in human health monitoring. This work can be considered for publication after major revision.

Comments:

  1. The introduction part further can be improved by considering the following references:  10.1038/s41598-021-95977-6; 10.1016/j.snb.2022.131507; 10.1016/j.sna.2015.03.003.
  2. Figure 4 describes different salt solutions used for device measurements. However, I haven't found any information on salts' names and their representative humidity sensing levels.
  3. It is highly recommended to compare ZnO-based sensors with already reported works. The following parameters should be considered Sensitivity, humidity range, linearity/non linearity, sensor design type, and applications.
  4. How well does this humidity sensor perform at different temperatures 25 (room temp), 40, 50 so on to 100 degrees centigrade? 
  5.  The author should add humidity sensor stability using salt solutions.
  6. Author revise english grammar and proofread thoroughly. 

Round 2

Reviewer 3 Report

Reviewer Comments:

The author has revised the manuscript and provided all missing information. At current this can be considered for publication in micromachines. 

This manuscript is a resubmission of an earlier submission. The following is a list of the peer review reports and author responses from that submission.